# Dendritic NMDA receptors in parvalbumin neurons enable strong and stable neuronal assemblies

Jonathan H Cornford[1†]*, Marion S Mercier[1†], Marco Leite[1], Vincent Magloire[1], Michael Häusser[2], Dimitri M Kullmann[1]*

[1]UCL Queen Square Institute of Neurology, University College London, London, United Kingdom; [2]Wolfson Institute for Biomedical Research, University College London, London, United Kingdom

**Abstract** Parvalbumin-expressing (PV+) GABAergic interneurons mediate feedforward and feedback inhibition and have a key role in gamma oscillations and information processing. The importance of fast synaptic recruitment and action potential initiation and repolarization, and rapid synchronous GABA release by PV+ cells, is well established. In contrast, the functional significance of PV+ cell NMDA receptors (NMDARs), which generate relatively slow postsynaptic currents, is unclear. Underlining their potential importance, several studies implicate PV+ cell NMDAR disruption in impaired network function and circuit pathologies. Here, we show that dendritic NMDARs underlie supralinear integration of feedback excitation from local pyramidal neurons onto mouse CA1 PV+ cells. Furthermore, by incorporating NMDARs at feedback connections onto PV+ cells in spiking networks, we show that these receptors enable cooperative recruitment of PV+ interneurons, strengthening and stabilising principal cell assemblies. Failure of this phenomenon provides a parsimonious explanation for cognitive and sensory gating deficits in pathologies with impaired PV+ NMDAR signalling.
DOI: https://doi.org/10.7554/eLife.49872.001

*For correspondence:
jhcornford@gmail.com (JHC);
d.kullmann@ucl.ac.uk (DMK)

†These authors contributed equally to this work

**Competing interests:** The authors declare that no competing interests exist.

## Introduction

Interactions among cell assemblies underlie information representation and processing in the brain (*Buzsáki, 2010*). Inhibitory interneurons, including fast-spiking PV+ cells, which mediate feedforward and feedback inhibition and are central to gamma oscillations, have a major role in segregating excitatory principal cells into functional groups. PV+ cells have broad receptive fields inherited from multiple converging heterogeneously tuned principal neurons (*Kerlin et al., 2010*) and coupled with their powerful somatic inhibition of principal cells, they are positioned to mediate a 'winner-takes-all' scheme in which neuronal assemblies inhibit each other (*Agetsuma et al., 2018*; *Trouche et al., 2016*).

The biophysical properties of PV+ cells that make them suited to fast inhibition of target neurons are well established (*Jonas et al., 2004*). These properties are critical for functions such as the enforcement of narrow temporal integration, input normalization, and sparsification of neuronal assemblies (*de Almeida et al., 2009*; *Pouille et al., 2009*; *Pouille and Scanziani, 2001*). However, PV+ interneurons are also equipped with NMDARs whose slow kinetics and nonlinear voltage dependence do not appear well-aligned with fast inhibition of principal cells. Although NMDARs contribute relatively less to synaptic excitation of PV+ cells than principal neurons (*Geiger et al., 1997*; *Lamsa et al., 2007*; *Matta et al., 2013*), several sources of evidence suggest that they are important for the normal operation of cell assemblies. In particular, genetic deletion of NMDARs in PV+ interneurons disrupts both gamma rhythms (*Carlén et al., 2012*) and spatial representations

(*Korotkova et al., 2010*). Moreover, impaired NMDAR-mediated signaling in PV+ interneurons has been suggested to be a core feature of schizophrenia (*Coyle, 2012*; *Lisman et al., 2008*). Indeed, genetic manipulation of the schizophrenia risk genes encoding neuregulin and ErbB4, which amongst other functions regulate NMDARs, impairs recruitment of PV+ interneurons and recapitulates some features of the disease (*Del Pino et al., 2013*; *Kotzadimitriou et al., 2018*).

A recent study investigating plasticity rules of glutamatergic inputs onto CA1 PV+ interneurons reported NMDAR-dependent long-term potentiation (LTP) at feedback synapses from local pyramidal neurons but not at feedforward connections made by Schaffer collaterals (*Le Roux et al., 2013*), and attributed the difference to a larger NMDAR conductance at feedback synapses. A natural question prompted by these findings is the degree to which NMDARs contribute to synaptic integration of glutamatergic feedback inputs on PV+ cells. In principal neurons, NMDAR-mediated dendritic nonlinearities enhance the computing capacity of individual cells (*Gasparini and Magee, 2006*; *Losonczy and Magee, 2006*; *Poirazi and Mel, 2001*; *Stuart and Spruston, 2015*). Do NMDARs have an analogous function in PV+ interneurons? Furthermore, given the importance of excitatory feedback connections on interneurons for microcircuit motifs, how do NMDARs in PV+ interneurons affect interactions between neuronal assemblies?

Here we combine in vitro optogenetic stimulation and two-photon glutamate uncaging with modeling to assess the role of NMDARs at excitatory feedback connections onto mouse hippocampal CA1 PV+ interneurons. We show that NMDARs at feedback synapses mediate integrative dendritic nonlinearities in PV+ interneurons. Importantly, this mechanism can be exploited to promote the formation of robust cell assemblies that are stable in the face of distracting noise.

## Results

### Differential input integration at stratum oriens and stratum radiatum dendrites of PV+ interneurons

Experiments were performed in acute hippocampal slices from mice obtained by crossing PV-Cre mice with Ai9 mice, and tdTomato expression was used to target fast-spiking PV+ interneurons in CA1 stratum pyramidale. Such neurons, which mainly comprise basket cells in addition to axo-axonic and bistratified cells (*Bezaire and Soltesz, 2013*), receive excitatory feedforward inputs across the full extent of their dendritic trees, in both strata radiatum and oriens. In contrast, feedback inputs from axon collaterals of local pyramidal cells are confined to dendrites in the stratum oriens (*Amaral et al., 1991*). In order to compare the contribution of NMDARs to dendritic integration of feedforward and feedback excitatory postsynaptic potentials (EPSPs), we took advantage of the anatomical restriction of feedback inputs onto oriens dendrites, and recorded somatic responses to two-photon glutamate uncaging at multiple sites within a 15 μm dendritic segment in either stratum oriens or stratum radiatum (*Figure 1A*). Activation of individual uncaging locations in either stratum evoked uncaging-evoked EPSPs (uEPSPs) that were comparable in amplitude and kinetics to spontaneous EPSPs (*Figure 1B* and *Figure 1—figure supplement 1*), consistent with a high density of excitatory synapses innervating PV+ interneuron dendrites (*Gulyás et al., 1999*). To quantify the degree of nonlinearity of dendritic integration, we compared compound uEPSPs elicited by near-synchronous activation of increasing numbers of uncaging locations to the arithmetic sum of individual uEPSPs at the same sites (*Figure 1C*). Activation of sites on dendrites in stratum oriens revealed supralinear uEPSP summation (peak amplitude nonlinearity: 24.0 ± 4.5%, mean ± SEM, n = 14; *Figure 1D,E*; unscaled responses in *Figure 1—figure supplement 2*). This nonlinearity was even larger when measured using the time-integral of uEPSPs measured between 0 and 50 ms from onset (time-integral nonlinearity: 54.0 ± 10.1%; *Figure 1F*). In contrast, when glutamate was uncaged along dendritic segments in stratum radiatum, uEPSPs summated in an approximately linear fashion (peak amplitude nonlinearity: 3.8 ± 5.0%, time-integral nonlinearity: 6.3 ± 7.6%, n = 9; oriens vs. radiatum p=0.0083 and p=0.0028 for peak amplitude and time-integral comparisons respectively, unpaired *t*-tests, *Figure 1D–F*). The difference between strata was also observed in a subset of paired recordings in which dendrites in both strata were tested (*Figure 1—figure supplement 3*). There was no consistent relationship between integration nonlinearity and either uncaging distance from soma or the size of the arithmetic sum of the uEPSPs (*Figure 1—figure supplement 4*). Given that synapses in stratum oriens are innervated by both local pyramidal neurons and Schaffer collaterals, the striking

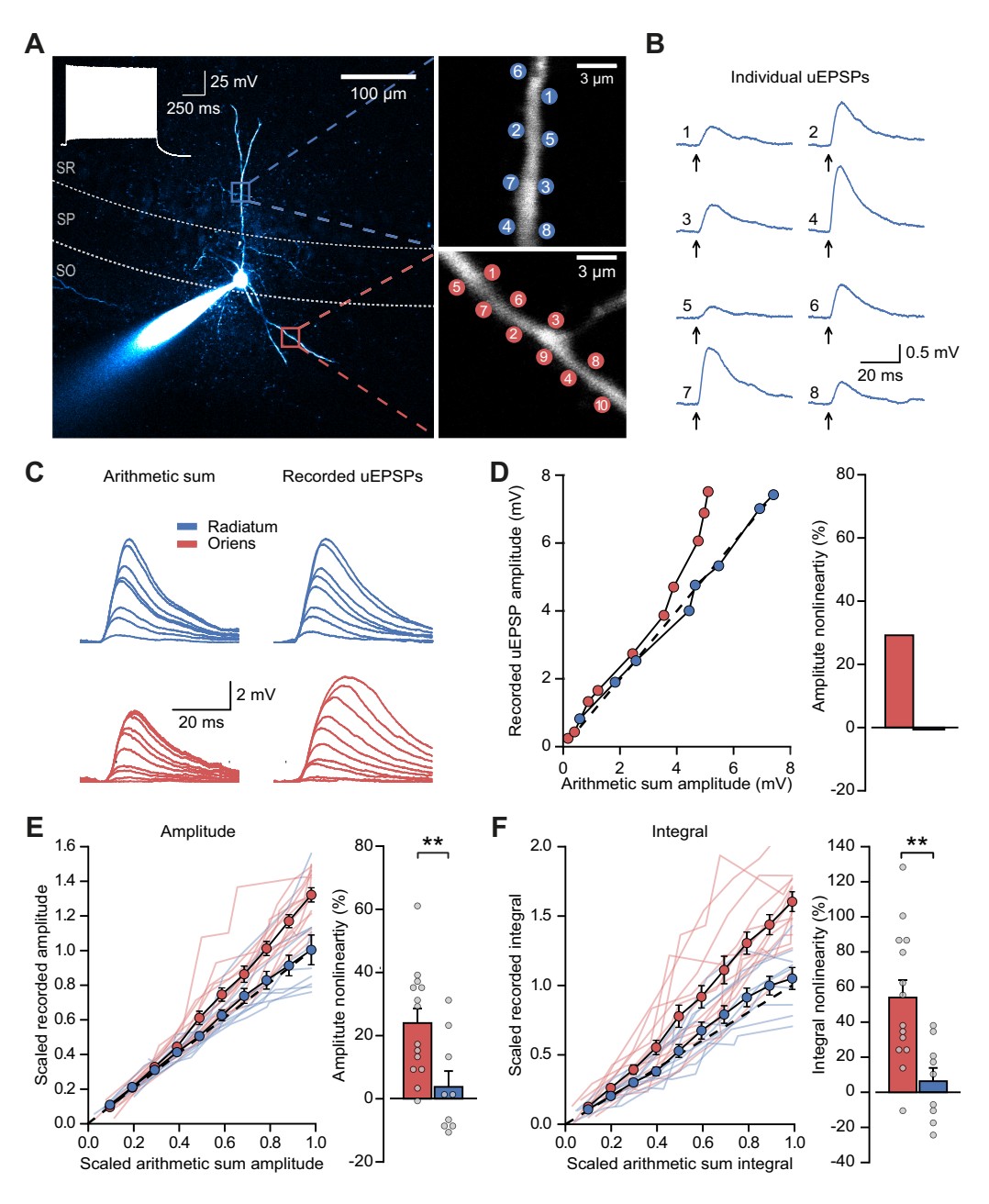

**Figure 1.** Differential input integration at stratum oriens and stratum radiatum dendrites of PV+ interneurons. (**A**) Two-photon z-projection image of a PV+ interneuron recorded via a patch pipette in stratum pyramidale (SP) and filled with Alexa-594 (left, inset: firing pattern in response to current injection), with two dendritic regions of interest at higher magnification (right: top, stratum radiatum, SR; bottom, stratum oriens, SO), showing glutamate uncaging locations (numbered). (**B**) Individual uEPSP responses from radiatum dendritic locations shown in (**A**). (**C**) Comparison of arithmetic sum of individual uEPSPs and recorded uEPSPs evoked by near-synchronous uncaging at multiple locations in stratum radiatum (blue) and oriens (red). (**D**) Peak amplitudes of recorded uEPSPs plotted against arithmetically summed waveforms for the two regions shown in (**A**). Dashed line shows line of identity. Right: bar chart showing percentage amplitude nonlinearity. Red: oriens, blue: radiatum. (**E**) Summary of scaled peak amplitude comparisons for all cells (oriens locations: n = 14, radiatum locations: n = 9). Filled circles and error bars indicate mean ± SEM. Right: bar chart showing quantification of amplitude nonlinearity. (**F**) Time-integral nonlinearity plotted as for (**E**). **: p<0.01.

DOI: https://doi.org/10.7554/eLife.49872.002

The following figure supplements are available for figure 1:

*Figure 1 continued on next page*

*Figure 1 continued*

**Figure supplement 1.** Somatic glutamate uncaging-evoked membrane responses.
DOI: https://doi.org/10.7554/eLife.49872.003
**Figure supplement 2.** Unscaled uEPSP integration location-dependent nonlinearity.
DOI: https://doi.org/10.7554/eLife.49872.004
**Figure supplement 3.** uEPSP integration location-dependent nonlinearity by cell.
DOI: https://doi.org/10.7554/eLife.49872.005
**Figure supplement 4.** uEPSP nonlinearity does not depend on uncaging location distance from soma or on the size of the arithmetic sum of uEPSPs.
DOI: https://doi.org/10.7554/eLife.49872.006
**Figure supplement 5.** Compound uEPSPs from uncaging locations clustered on a single dendrite display larger nonlinearities than when distributed across two dendrites.
DOI: https://doi.org/10.7554/eLife.49872.007

supralinear summation of uEPSPs uncovered here may underestimate the true extent of dendritic nonlinearity at feedback connections.

We repeated these experiments with uncaging locations distributed across two dendrites in stratum oriens, and compared the degree of non-linearity to that observed when uncaging was confined to either one of the dendrites. The degree of supralinear summation was significantly lower when uncaging was distributed across two dendrites (peak amplitude nonlinearity, 39.3 ± 11.7% vs 16.5 ± 5.6% for within-dendrite and across-dendrite uncaging respectively, p=0.016, n = 12; paired *t*-test; *Figure 1—figure supplement 5*). This result indicates that the spatial clustering, or conversely dispersion, of co-active dendritic inputs to PV+ interneurons has an important role in input integration.

## NMDAR expression and dendrite morphology underlie stratum-dependent differences in synaptic integration

Supralinear dendritic integration in pyramidal neurons depends on the recruitment of voltage-dependent conductances. We therefore investigated the role of such conductances in PV+ interneurons. In line with previous evidence for a substantial NMDAR component at feedback inputs onto PV+ cells (*Le Roux et al., 2013*), supralinear dendritic summation in stratum oriens was abolished when NMDARs were blocked by D-AP5 (100 µM) (time-integral nonlinearity: 2.5 ± 3.0%, vs control without the drug p=0.0004, n = 10; *Figure 2A–C*). Dendritic integration in stratum radiatum was unchanged from control conditions (time-integral nonlinearity: 3.3 ± 2.6%, vs control p=0.88, n = 4; *Figure 2A–C*). In contrast to D-AP5, the sodium channel blocker tetrodotoxin (TTX, 100 nM) did not significantly affect integration in either stratum oriens or radiatum (oriens time-integral nonlinearity 40.1 ± 5.6%, vs control p=0.23, n = 16; radiatum time-integral nonlinearity 9.4 ± 3.3%, vs control p=0.71, n = 9; *Figure 2D*). The failure of TTX to affect uEPSP integration is consistent with the view that PV+ interneuron dendrites generally do not support regenerative events (*Hu et al., 2010*) (although see *Chiovini et al., 2014*). The effects of pharmacological manipulations were consistent whether measuring time-integrals or peak uEPSP amplitudes (*Figure 2—figure supplement 1*). Furthermore, uncaging distances from soma were comparable across all conditions, as were somatic uEPSP amplitudes (*Figure 2—figure supplement 2*). Dendrites of PV+ interneurons that mediate feedback inhibition, but not those mediating purely feedforward inhibition, thus exhibit NMDAR-dependent supralinear input integration. These findings imply that clusters of coactive synapses supplied by local pyramidal neurons cooperate via depolarization-dependent relief of NMDARs from $Mg^{2+}$ blockade.

The results above are consistent with previous evidence of a larger NMDAR/AMPAR conductance ratio at feedback than feedforward synapses, estimated by electrically stimulating axons in the alveus/stratum oriens or stratum radiatum respectively, while clamping PV+ interneurons at positive and negative potentials to separate AMPAR and NMDAR components (*Le Roux et al., 2013*; *Pouille and Scanziani, 2004*). However, the low input resistance of PV+ interneurons, together with different dendritic morphologies in strata oriens and radiatum (*Hu et al., 2010*), potentially confounds the comparison of excitatory postsynaptic currents (EPSCs) originating from the two locations and recorded at positive holding potentials (*Williams and Mitchell, 2008*). We therefore used an alternative experimental design to estimate the relative contribution of AMPARs and NMDARs.

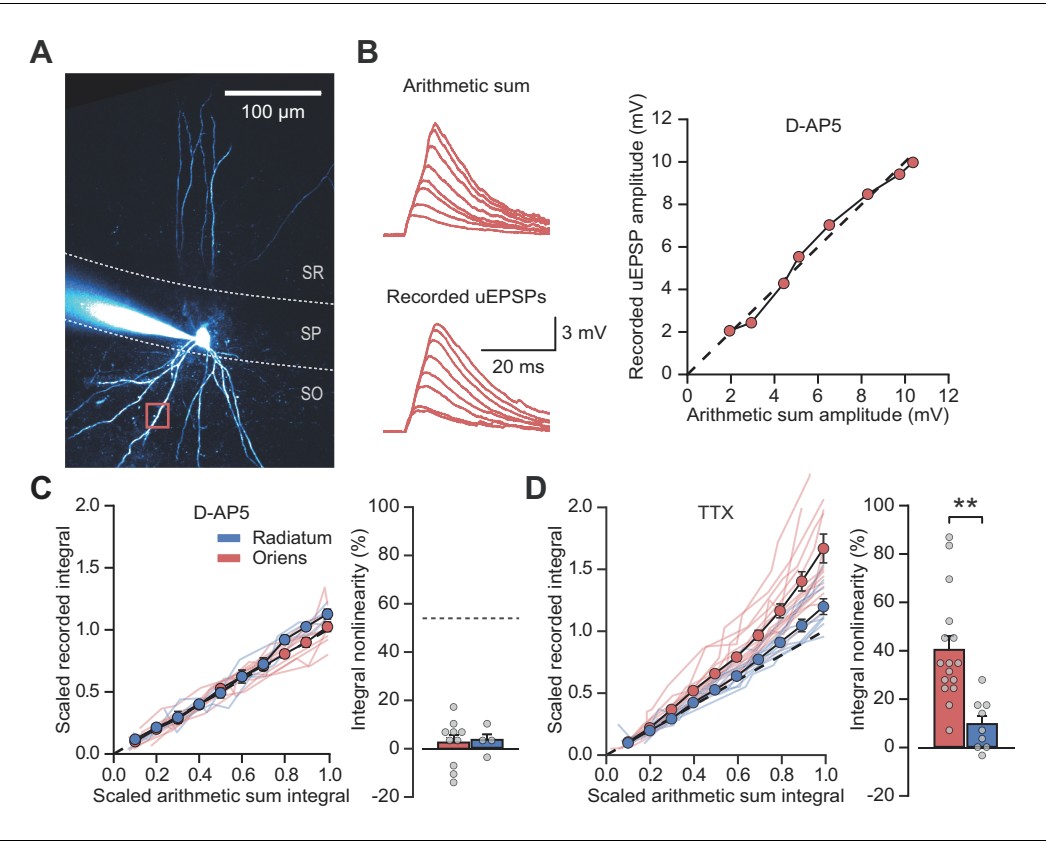

**Figure 2.** NMDARs mediate stratum oriens dendrite synaptic integration supralinearity. (**A**) Two-photon z-stack of PV+ interneuron in CA1 region of hippocampus. Red box marks glutamate uncaging location. (**B**) Comparison of arithmetic and recorded uEPSP summation waveforms in the presence of D-AP5. Right: peak recorded amplitude vs peak arithmetic amplitude. (**C**) Summary data of time-integrals plotted against arithmetic sum time-integrals for 14 dendritic locations recorded in D-AP5 (oriens locations: n = 10, radiatum locations: n = 4). Right: quantified synaptic integration nonlinearity. The dashed line marks the average magnitude of oriens nonlinearity from *Figure 1F*. (**D**) Summary data for 25 dendritic locations recorded in TTX (oriens locations: n = 16, radiatum locations: n = 9). Right: quantification of synaptic integration nonlinearity. Filled circles and error bars indicate mean ± SEM.

DOI: https://doi.org/10.7554/eLife.49872.008

The following figure supplements are available for figure 2:

**Figure supplement 1.** Peak amplitude nonlinearity in oriens dendrites is abolished by blocking NMDARs with D-AP5 but not by blocking sodium channels with TTX.
DOI: https://doi.org/10.7554/eLife.49872.009

**Figure supplement 2.** Arithmetic sum maximum uEPSP amplitudes, and integration nonlinearity vs uncaging location distances, across pharmacological conditions.
DOI: https://doi.org/10.7554/eLife.49872.010

---

Specifically, we recorded EPSCs in a low (0.1 mM) extracellular $[Mg^{2+}]$ solution to partially unblock NMDARs while holding PV+ interneurons at –60 mV, and used sequential addition of AMPAR and NMDAR blockers to separate the two components of transmission (*Figure 3A*).

Pharmacological dissection of EPSCs in 0.1 mM $[Mg^{2+}]$ revealed a > 2 fold greater NMDAR/AMPAR charge ratio when stimulating in the alveus than when stimulating in stratum radiatum (charge ratio: 3.5 ± 0.7 vs 1.3 ± 0.3, p=0.0017, n = 10, paired *t*-test; *Figure 3A*). The decay time constant of the NMDAR-mediated EPSCs was similar for the two inputs (Schaffer collaterals: 154.6 ± 25.9 ms vs alveus: 148.1 ± 13.4 ms, p=0.8, n = 10, paired *t*-test; *Figure 3B*) providing no evidence for differences in NR2B subunit inclusion, again consistent with previous work that showed similar effects of selective blockade of NR2B-containing receptors (*Le Roux et al., 2013*) (although see *Matta et al., 2013*).

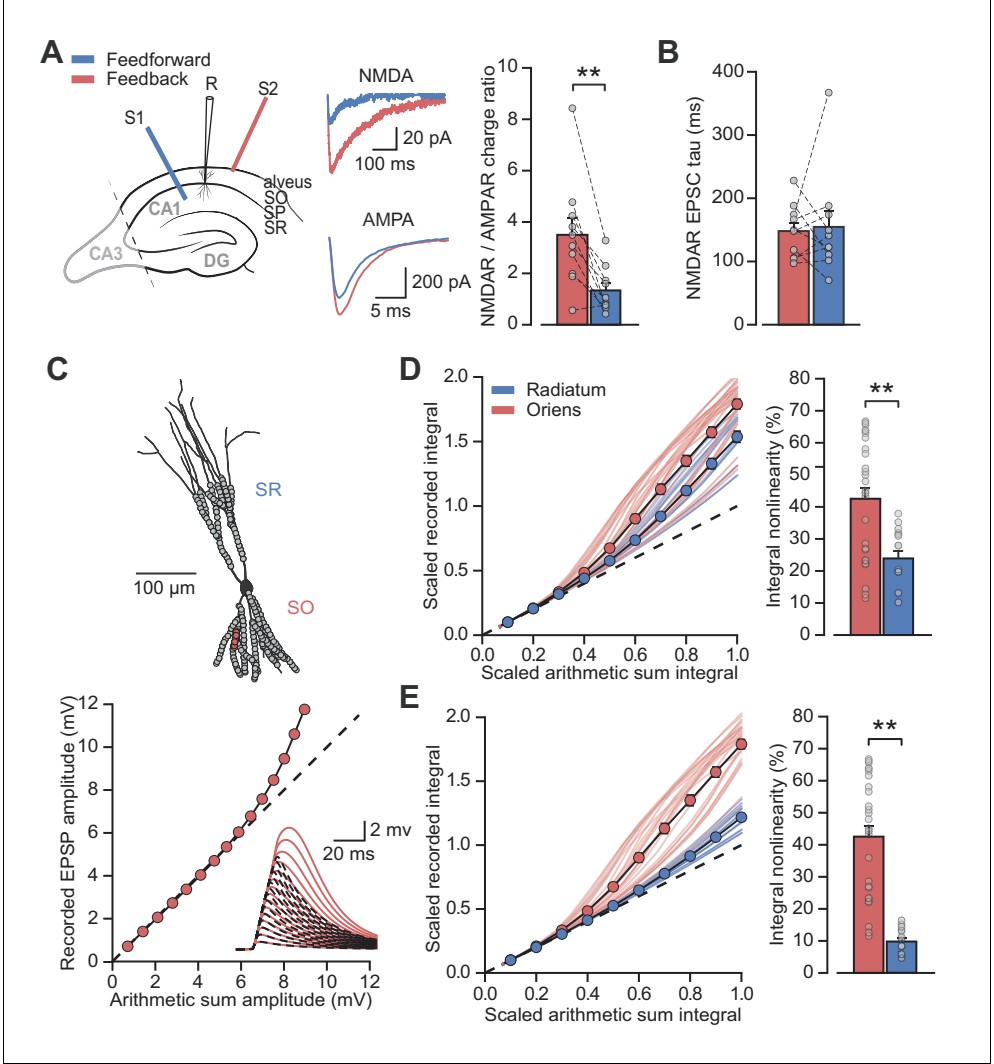

**Figure 3.** Differential NMDAR expression and dendrite morphology explain stratum-dependent synaptic integration difference. (**A**) Schematic describing stimulation of feedforward (S1, blue) and antidromic stimulation of feedback (S2, red) axons. Middle: example paired AMPAR and NMDAR EPSC components in low [Mg$^{2+}$]. Right: NMDAR/AMPAR charge ratios (n = 10). (**B**) Decay time constants of NMDAR-mediated EPSCs recorded in the same PV+ neurons in response to stimulation of feedforward (blue) and feedback (red) axons (n = 10). (**C**) Top: reconstruction of a PV+ interneuron (axon not shown). Simulated synaptic locations are shown in gray. Bottom: example simulated uncaging experiment at the synapses marked with red circles; graph shows recorded EPSP amplitudes vs arithmetic sum of EPSP amplitudes. Inset: red solid lines, recorded summation; dashed black lines, arithmetic summation; waveforms calculated from individual synaptic responses. (**D**) Scaled recorded time-integrals vs scaled arithmetic sum of time-integrals at all locations with equal NMDAR conductance (oriens locations: n = 28, radiatum locations: n = 16). Right: quantified synaptic integration nonlinearity. (**E**) As (**D**), but with reduced NMDAR/AMPAR conductance ratio at radiatum dendrites. Oriens data replotted from (**D**).

DOI: https://doi.org/10.7554/eLife.49872.011

The following figure supplement is available for figure 3:

**Figure supplement 1.** Simulations including polyamine modulation of AMPARs show synaptic integration differences between strata oriens and radiatum dendrite locations.

DOI: https://doi.org/10.7554/eLife.49872.012

Dendrites of PV+ interneurons in stratum oriens are generally thinner and shorter than those in stratum radiatum (*Gulyás et al., 1999*) suggestive of a higher effective local input impedance. This raises the possibility that, in addition to enriched NMDAR expression, oriens dendrites may be depolarized more effectively by glutamate uncaging resulting in an enhanced relief of NMDARs

from voltage-dependent Mg$^{2+}$ block (*Branco et al., 2010*). To investigate the relationship between synaptic integration and dendritic geometry we used a detailed compartmental model of a CA1 PV+ interneuron (*Figure 3C*). Voltage-dependent conductance densities and membrane properties were implemented according to previously published models (*Hu and Jonas, 2014*; *Nörenberg et al., 2010*), and the relative densities of synaptic AMPARs and NMDARs were initially assumed to be the same on oriens and radiatum dendrites. Simulation parameters closely followed the uncaging experiments, with clusters of synapses activated across the range of experimentally measured locations. These simulations revealed supralinear summation of EPSPs recorded at the soma that was more pronounced for stratum oriens than for stratum radiatum dendrites (oriens vs radiatum time-integral nonlinearity: 42.5 ± 3.5% vs 23.9 ± 2.3%, p<0.001, *Figure 3D*), supporting a role for dendritic morphology in mediating the difference between strata. The simulation results were very similar whether AMPARs were assumed to show polyamine-dependent inward rectification or to have a fixed open-channel conductance (*Figure 3—figure supplement 1*). The ~2 fold difference in supralinearity between strata was, however, smaller than the >8 fold difference observed experimentally (oriens vs radiatum time-integral nonlinearity: 54.0 ± 10.1% vs 6.3 ± 7.6%; *Figure 1*).

Reducing the simulated NMDAR/AMPAR conductance ratio at radiatum dendrites to half that of the oriens dendrites, in line with results from experiments in *Figure 3A*, improved agreement with the glutamate uncaging data (time-integral supralinearity in simulations: 42.5 ± 3.5% vs 8.5 ± 0.8% for stratum oriens vs stratum radiatum; *Figure 3E*). The difference in dendritic integration in oriens and radiatum dendrites observed experimentally (*Figure 1*) may therefore be accounted for by a combination of differential NMDAR expression (greater in stratum oriens) and dendritic morphology (greater impedance in stratum oriens, thus facilitating depolarization and NMDAR opening).

## NMDAR recruitment at CA1 pyramidal cell feedback connections onto PV+ interneurons

While electrical stimulation of the alveus recruits local pyramidal cell axon collaterals, it may also recruit other extrinsic afferents that could contribute to the observed NMDAR currents. In order to isolate the feedback input from CA1 pyramidal cells to PV+ interneurons, and to measure the magnitude of the NMDAR component at these synapses, we combined voltage clamp using a Cs$^+$-based pipette solution with optogenetic stimulation of feedback fibers. ChR2 was selectively expressed in CA1 pyramidal cells by injecting an adeno-associated virus (AAV) encoding ChR2-EYFP under the control of the CaMKII promoter in CA1 of the dorsal hippocampus. We routinely verified that expression was confined to CA1 and did not spread to CA3 (*Figure 4A*). Wide-field illumination pulses (1 ms) of 470 nm light elicited monophasic EPSCs, in agreement with the low associative connectivity among CA1 pyramidal neurons (*Amaral et al., 1991*; *Deuchars and Thomson, 1996*) (*Figure 4B*). AMPAR- and NMDAR-mediated light-evoked feedback EPSCs were recorded at −60 mV and +60 mV, respectively, and NBQX was added to isolate the NMDAR component (*Figure 4B*). This revealed large NMDAR currents (amplitude: 459.1 ± 89.2 pA, integral: 39.1 ± 6.8 nA ms) and NMDAR/AMPAR ratios (amplitude: 0.7 ± 0.2, integral: 2.7 ± 0.8, *Figure 4C*), confirming abundant expression of functional NMDARs at feedback excitatory synapses on PV+ interneurons.

Our results so far argue that NMDARs mediate supralinear integration of uncaging-evoked responses in stratum oriens dendrites, and that synapses mediating feedback excitation of stratum oriens dendrites are enriched with NMDARs. Glutamate uncaging does not necessarily restrict NMDAR activation to synaptic receptors, leaving uncertain whether feedback innervation of PV+ interneurons is able to engage NMDARs under more physiological conditions. We therefore used the same optogenetic strategy, to ask whether feedback inputs from local CA1 pyramidal cells depolarize PV+ interneurons sufficiently to recruit NMDARs. We measured the contribution of NMDARs to optogenetically evoked EPSPs recorded in current clamp, whilst incrementing the light intensity through a duty cycle (*Figure 4D*). Perfusion of the NMDAR blocker D-AP5 significantly reduced both the average time-integral and the ratio of integral to peak of EPSPs evoked at the maximal light intensity (p=0.033; p=0.014; n = 5), but not the peak EPSPs (*Figure 4E,F*). Furthermore, in line with cooperative postsynaptic voltage-dependent relief of Mg$^{2+}$ blockade, we observed a larger NMDAR contribution to EPSPs elicited by stronger light pulses (*Figure 4G*). Together, these experiments confirm that synaptic glutamate release at feedback inputs from CA1 pyramidal cells can elicit NMDAR-mediated depolarization of PV+ interneurons.

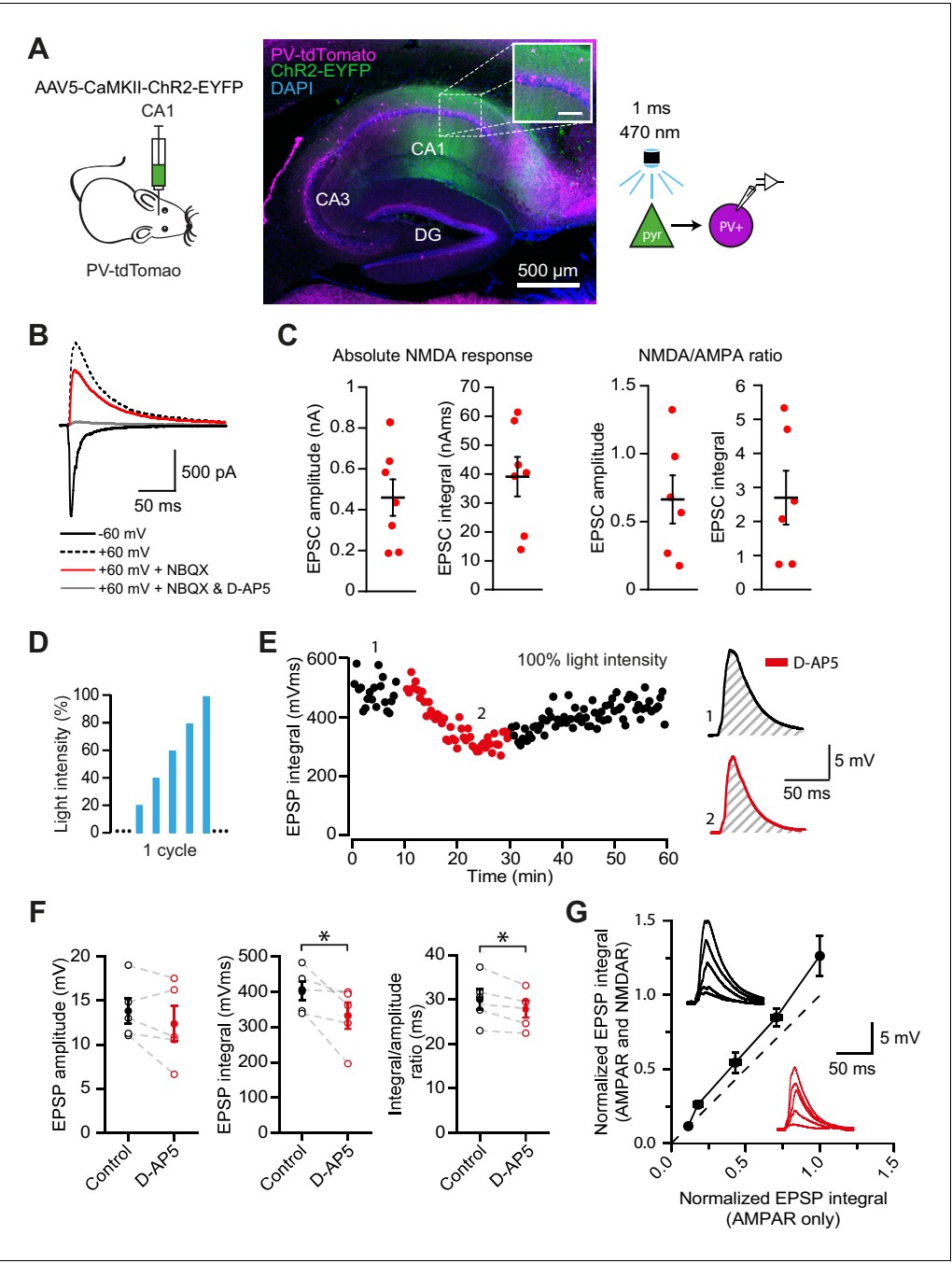

**Figure 4.** NMDAR recruitment at CA1 pyramidal cell feedback connections onto PV+ interneurons. (**A**) Schematic of viral injections into dorsal CA1 of PV-tdTomato mice (left), and confocal image of a sagittal hippocampal slice showing selective ChR2-EYFP expression in CA1 pyramidal cells (middle, inset scale: 100 μm). Right: schematic of optogenetic patch clamp experiments. (**B**) Example traces of light-evoked feedback EPSCs in a PV+ interneuron held at −60 mV (black), +60 mV (black dashed), +60 mV with application of NBQX (red) or +60 mV with NBQX and D-AP5 in voltage clamp. (**C**) Quantification of absolute NMDAR-mediated feedback EPSCs (amplitude and integral, left, n = 7) and NMDAR/AMPAR ratios (amplitude and integral, right, n = 6), measured from voltage clamp experiments as in (**B**). Black bars indicate mean ± SEM. (**D**) Schematic of optogenetic stimulation protocol for current clamp experiments: light power was cycled from 20% to 100% of power for maximal response (see Materials and methods). (**E**) EPSP integral of maximal response over time, with 20 min application of D-AP5 (red). (**F**) EPSP amplitude, EPSP integral and integral/amplitude ratio in the presence (red) or absence (black) of D-AP5 (n = 5, one-tailed *t*-tests; control = average of baseline and wash). Filled circles and error bars indicate

*Figure 4 continued on next page*

*Figure 4 continued*

mean ± SEM. (**G**) Normalized EPSP integrals (black example traces) vs normalized EPSP integrals in the presence of D-AP5 (red example traces), for all stimulation intensities (n = 5).
DOI: https://doi.org/10.7554/eLife.49872.013

## NMDAR recruitment at feedback connections onto PV+ interneurons strengthens and stabilizes neuronal assemblies

NMDAR-dependent supralinear dendritic integration increases the computational capacity of principal neurons (*Mel, 1992*). In an extreme case cooperativity among synapses could implement an AND gate where a somatic depolarization is conditional on more than one near-simultaneous excitatory input impinging on a dendritic branch. We asked how supralinear summation of feedback excitation of PV+ interneurons could affect local circuit behavior. Feedback recruitment of interneurons has been implicated in lateral inhibition, implementing a winner-takes-all mechanism (*de Almeida et al., 2009*). We therefore simulated a network of 250 excitatory point neurons (*Izhikevich, 2003*) reciprocally connected to a single fast-spiking inhibitory neuron (*Ferguson et al., 2014*; *Ferguson et al., 2013*) (*Figure 5A,B*). The inhibitory neuron received dual-component (AMPAR and NMDAR) synaptic conductances from the excitatory neurons (*Figure 5C*), and synapses located close to one another were allowed to interact cooperatively and engage the non-Ohmic behavior of NMDARs. The strength of interaction between individual excitatory synapses on the interneuron fell off with distance in an abstract input space (*Figure 5—figure supplement 1*), inspired by the experimental evidence for clustering of homotopic inputs on dendritic segments in principal neurons (*Iacaruso et al., 2017*; *Wilson et al., 2016*). Excitatory neurons were assumed to be driven by a Poisson process (denoted 'external drive') whose rate across the population was defined to be either clustered or dispersed in the input space. The PV+ interneuron also received a constant fraction of the external drive received by the principal neurons. The overall intensity of this drive was set such that the network entered a sparsely firing oscillatory state akin to a cortical gamma rhythm, with the simulated PV+ interneuron firing one-to-one with the gamma cycle. Excitatory neurons driven by a compact 'hump' of excitation in input space cooperated in recruiting NMDARs on the interneuron to a greater extent than equivalent excitation shuffled randomly in input space (*Figure 5D*). The disproportionate NMDAR activation by compact versus distributed excitation recapitulates multiple co-active synapses within a small region of the dendritic tree cooperating to relieve NMDARs from $Mg^{2+}$ blockade. Although individual pyramidal neurons fired sparsely, an effect of the gamma oscillation was to synchronize them so that the local depolarization was maximized.

Recruitment of NMDAR conductances in the interneuron also maintained sparse principal cell firing over several oscillatory cycles (*Figure 5—figure supplement 2*). In contrast, without NMDARs, the hump of active principal cells broadened as the oscillation stabilized. Principal cells at the core of the hump of activity (as defined by synaptic location on the interneuron dendritic tree) thus preferentially influence the firing of the interneuron as a result of the recruitment of NMDARs. We propose, therefore, that NMDARs on PV+ interneurons contribute to maintaining a sharp assembly representation, dependent on the spatial arrangement of active synapses on the dendritic tree of the interneuron.

Next, we simulated two similar networks mutually inhibiting one another (*Geisler et al., 2007*; *Trouche et al., 2016*) to understand how NMDARs in the inhibitory neurons could affect competition among cell assemblies. When one network received a stable and compact hump of excitation (again, with input space defined by location on the interneuron dendritic tree) it was much more likely to 'win' than a competing network receiving an equal amount of excitation that was dispersed (*Figure 6A*). This effect results from the additional interneuron depolarization mediated by NMDARs which were recruited by clustered synapses. The tendency for the network receiving a hump of excitation to win disappeared when NMDARs were removed from the inhibitory neurons (*Figure 6B,C*).

Finally, we explored the ability of the combined network to 'lock' onto one of two inputs of similar strength and compactness presented to the two sub-networks (*Figure 7*). An ethologically relevant analogous task in humans is the ability to stabilize perception of a Necker cube (*Figure 7A*). Although we make no claim as to how this task is solved, it exemplifies a situation where two sensory or cognitive representations compete for recruitment of a network. The net excitatory external drive

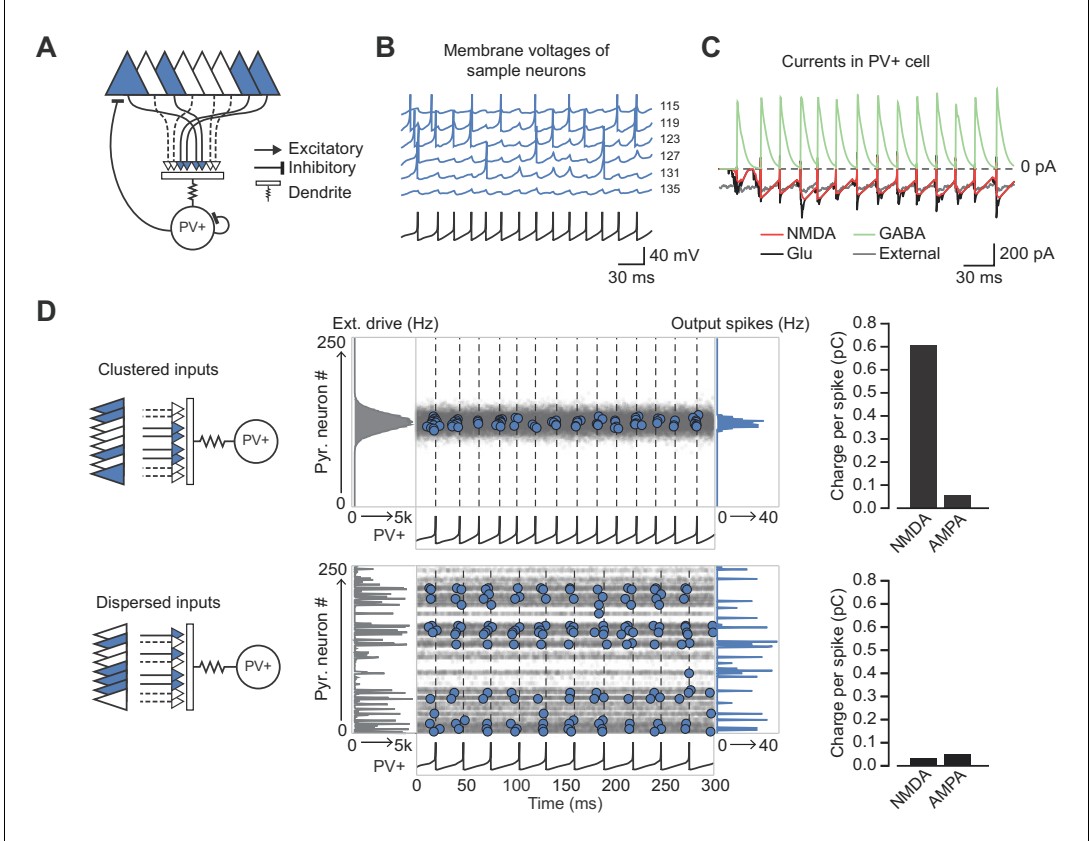

**Figure 5.** Network architecture and NMDAR recruitment at feedback connections. (**A**) Schematic of network structure. (**B**) Voltage traces of interneuron (black) and principal cells (blue, cell # at right) during network simulation. The network was driven by an asynchronous barrage of spikes, maximal in cell #125 ('clustered' input). (**C**) Corresponding currents in interneuron. Red: NMDAR currents from principal cells; green: GABAR currents from autaptic PV+ cell connections; black: sum of NMDAR and AMPAR currents from principal cells; gray: AMPAR currents from external drive. (**D**) Left: schematic showing cell assemblies receiving clustered (top) or dispersed (bottom) external inputs, and middle: corresponding summary plots of network simulation showing external drive input distribution (gray), pyramidal cell firing (blue, circles), and interneuron firing (black, and vertical dashed lines). Right: average NMDAR and AMPAR charge in interneuron per principal neuron spike. (Autaptic and feedback connections from PV+ cells are omitted from the schematic for clarity.).

DOI: https://doi.org/10.7554/eLife.49872.014

The following figure supplements are available for figure 5:

**Figure supplement 1.** Modeling principal cell input cooperation onto feedback interneurons.
DOI: https://doi.org/10.7554/eLife.49872.015

**Figure supplement 2.** NMDARs help to maintain a sparse and sharp representation of a 'hump' of excitation to the feedback circuit shown in *Figure 5D*.
DOI: https://doi.org/10.7554/eLife.49872.016

to each of the sub-networks was allowed to fluctuate independently with time (*Figure 7B*). As a result of this stochastic variability in the external drive strength, and neuronal accommodation, the combined network intermittently 'flipped' between the two inputs. However, the frequency of flipping increased steeply when the normalized conductance of NMDARs was decreased in the inhibitory neurons, resulting in a flickering of the dominant assemblies (*Figure 7C*). In contrast, the frequency of flipping was relatively unaffected by reducing the AMPAR conductance (*Figure 7D*). Because gamma oscillations in the hippocampus are nested within slower theta oscillations (*Chrobak and Buzsáki, 1998*), we repeated the simulations while modulating the external drive with a theta oscillation. This yielded qualitatively similar results (*Figure 7—figure supplement 1*). We thus conclude that NMDAR-mediated cooperative interactions among clustered synapses on an interneuron stabilize cell assemblies.

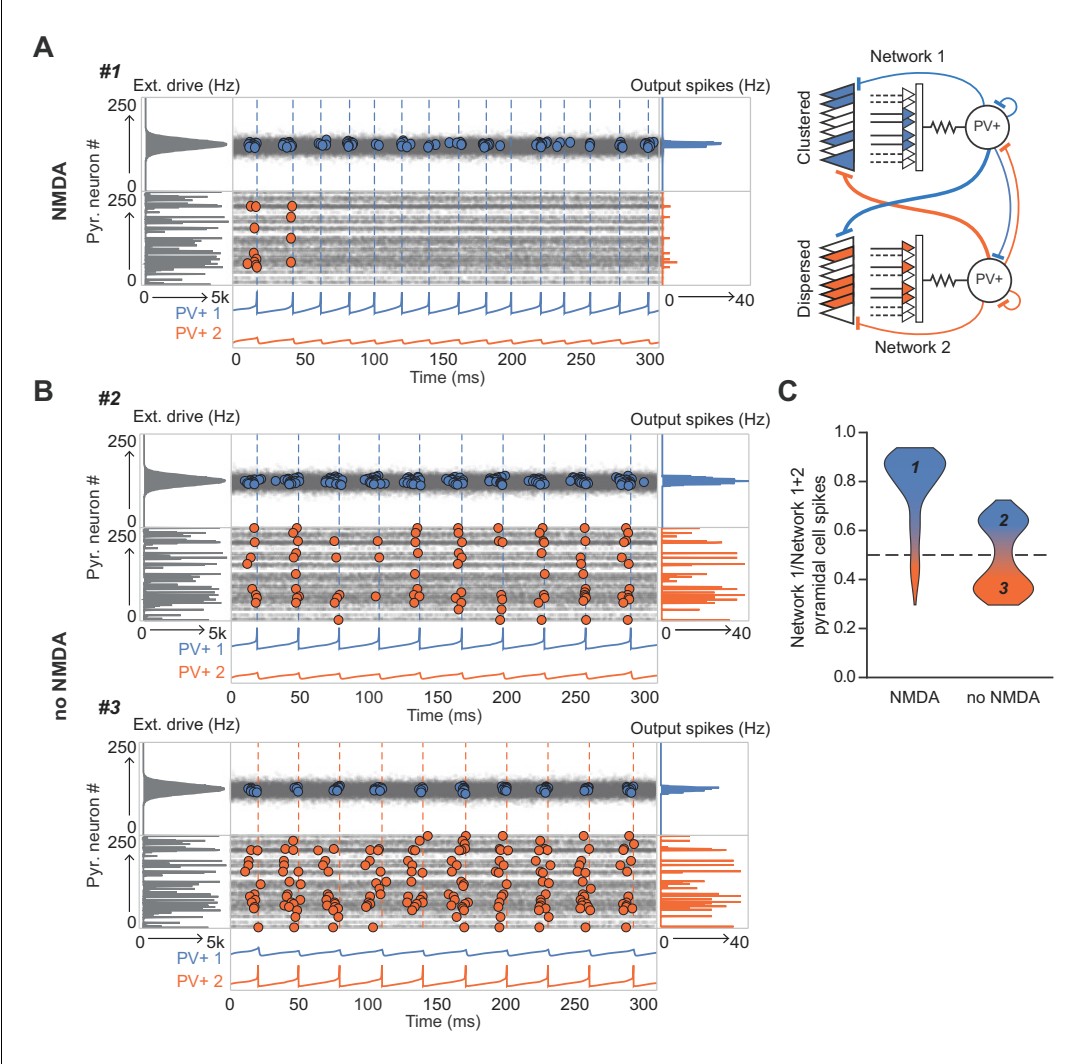

**Figure 6.** The role of NMDARs at feedback connections in cell assembly competition. (**A**) Right: schematic showing competing cell assemblies with clustered (blue) or dispersed (orange) inputs; left: example simulation of lateral inhibition between these subnetworks with NMDARs at feedback connections to interneuron (input distribution: gray; pyramidal cell firing: blue/orange, circles; interneuron firing: blue/orange spikes and vertical dashed lines). The network receiving the clustered input out-competed the network receiving the dispersed input. (**B**) Same as (**A**) but without NMDARs at feedback connections, showing, in one case the network receiving clustered input firing more than the network receiving dispersed input (top), and in the other case the network receiving dispersed input winning (bottom). (**C**) Summary of 250 simulations showing ratio of principal cell spikes for each subnetwork with and without NMDARs at feedback inputs onto interneuron. Numbers correspond to simulations illustrated in (**A**) and (**B**).
DOI: https://doi.org/10.7554/eLife.49872.017

## Discussion

The present study shows that clustered excitatory synapses on stratum oriens dendrites of CA1 PV+ interneurons interact supralinearly, challenging the view that they act as linear integrators of synaptic inputs (see also *Tzilivaki et al., 2019*). The high impedance of stratum oriens dendrites, which are innervated by axon collaterals of local pyramidal neurons, facilitates the cooperative recruitment of NMDARs. In addition, voltage clamp experiments show a larger NMDAR/AMPAR ratio at feedback inputs than at feedforward inputs. We place these results in the context of cell assembly competition by including nonlinear feedback integration in a spiking neural network model. NMDARs at synapses on a simulated feedback interneuron allow multiple principal cells co-innervating a subset of its input space, reminiscent of a dendritic branch, to interact cooperatively in recruiting the interneuron. We show that cooperative assemblies are more able to efficiently engage interneurons in the inhibition

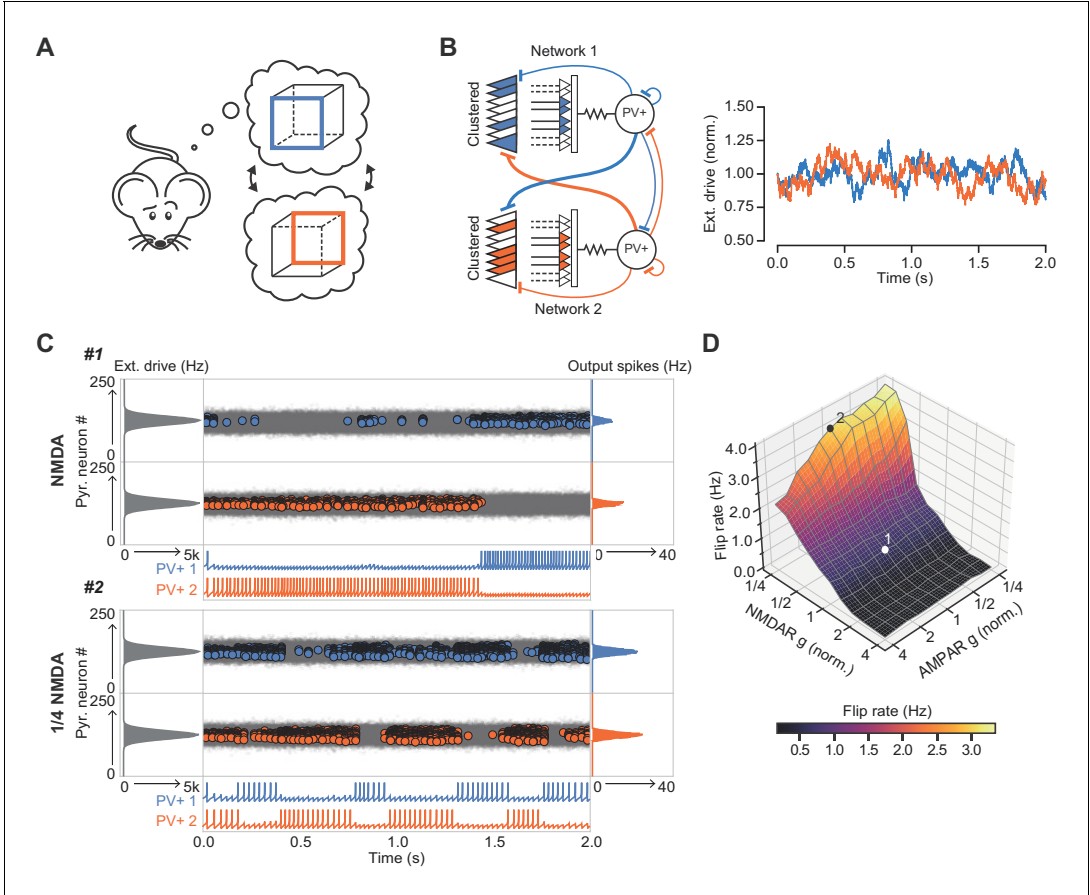

**Figure 7.** The role of NMDARs at feedback connections in cell assembly stability. (**A**) Cartoon illustrating a bistable neural representation. (**B**) Schematic of competing subnetworks both receiving clustered inputs (left) with random fluctuations in external input strength (plotted right). (**C**) Example simulation of network activity with NMDARs at synapses on interneurons (top), and with NMDARs scaled down to 25% of baseline (bottom). Input distribution: gray; pyramidal cell firing: blue/orange, circles; interneuron firing: blue/orange spikes. Although the dominant networks flipped spontaneously in both cases, the frequency of flipping was much higher with the down-scaled NMDARs. (**D**) Plot of network flip rate vs NMDAR and AMPAR conductance. White point (1): baseline NMDAR simulation parameters; black point (2): NMDARs down-scaled to 25%.
DOI: https://doi.org/10.7554/eLife.49872.018

The following figure supplement is available for figure 7:

**Figure supplement 1.** The role of NMDARs at feedback connections in cell assembly stability with theta-modulated external drive.
DOI: https://doi.org/10.7554/eLife.49872.019

of rival competing assembly representations. Furthermore, we report that inclusion of NMDAR conductances at feedback synapses stabilizes cell assemblies, allowing the network to 'lock' on to an input. An adaptive role of NMDARs in feedback excitation of PV+ interneurons can therefore be to facilitate the emergence of strong and stable cell assemblies.

Extrapolating from the behavior of a spiking neural network model to information processing clearly depends on a number of assumptions, not least that the principles underlying NMDAR-dependent input integration observed in CA1 PV+ interneurons apply generally throughout the brain, and that clustering of homotopic inputs on inhibitory dendritic segments obeys the same rules as in excitatory neurons (*Iacaruso et al., 2017*; *Wilson et al., 2016*). In addition, while PV+ cells predominantly show NMDAR-independent anti-Hebbian LTP (*Lamsa et al., 2007*), dependent on calcium-permeable AMPARs, feedback synapses also display NMDAR-dependent Hebbian plasticity rules (*Le Roux et al., 2013*). In this study, we isolate the integrative properties of NMDARs from their role in synaptic plasticity, but it is expected that plasticity mechanisms may also contribute to the network phenomena described here. For example, NMDAR-dependent plasticity, which could be induced by cooperative feedback synaptic integration, would allow PV+ interneurons to be wired

into stimulus-specific ensembles (*Khan et al., 2018*). Overall, therefore, NMDAR-dependent supra-linear integration in the feedback inhibitory loop potentially expands the computational power of a canonical cortical motif.

Selective knockdown of NMDARs in PV+ interneurons has previously been shown to cause a range of functional impairments, including working-memory deficits and a reduction in the precision of hippocampal spatial representations (*Korotkova et al., 2010*). The network simulations presented here provide a mechanistic explanation for some of these results, in particular, a reduction in spatial information conveyed by principal cell spiking (*Korotkova et al., 2010*). Other studies highlight a role for PV+ cells and inhibition in cell-assembly competition: in the visual cortex PV+ inhibition increases assembly similarity (*Agetsuma et al., 2018*) and in the hippocampus silencing of a dominant assembly was shown to uncover an alternative previously inhibited assembly (*Trouche et al., 2016*). The present study indicates that NMDARs may be integral to these functions.

Our network simulation results also resonate with multiple convergent findings that implicate PV+ cell NMDAR hypofunction at the centre of schizophrenia pathophysiology (*Bygrave et al., 2016*; *Lisman et al., 2008*; *Nakazawa et al., 2012*). For instance, NMDAR blockers have been shown to recapitulate some features of schizophrenia in healthy individuals (*Krystal et al., 1994*). When coupled with the observation that NMDAR antagonists cause a net disinhibition of principal cell activity (*Homayoun and Moghaddam, 2007*; *Jackson et al., 2004*), it has been suggested that cortical circuits are especially vulnerable to failure of NMDAR-mediated signaling in PV+ interneurons. Moreover, post-mortem studies have revealed a selective loss of PV+ interneurons in people with schizophrenia (*Lewis et al., 2012*) and overexpression of Neuregulin 1, a leading schizophrenia susceptibility gene, is associated with a reduction of NMDARs on PV+ cells (*Kotzadimitriou et al., 2018*). Destabilization of dominant neuronal assemblies, as shown here to result from impaired NMDAR signaling on PV+ interneurons, may thus explain a failure of sensory gating (*Javitt and Freedman, 2015*) and evidence for reduced cognitive control, for instance in the Necker cube test (*McBain et al., 2011*), in schizophrenia.

Feedback inhibition, such as that mediated by PV+ interneurons, is thought to be critical for preventing runaway excitation. Indeed, a failure of feedback inhibitory restraint is a major factor in the emergence of pathological states such as epileptic seizures. Given the importance of inhibition, how then do PV+ interneurons participate in assemblies that are composed of cells that they also regulate? One possibility is that PV+ cells are 'transient allies' (*Buzsáki, 2010*), only aligning with principal cell assemblies over short time windows and at specific times. For example, in a recent study of circuit changes in the visual cortex during learning, PV+ cells were found to become more selective to task-relevant stimuli by increasing their coupling to stimuli selective principal cells and becoming less influenced by the general activity of the remaining surrounding network (*Khan et al., 2018*).

It is worth noting that feedback interneurons operating over longer time scales, for example somatostatin-positive oriens-lacunosum/moleculare interneurons, or cholecystokinin-expressing basket cells, may also be suited to mediate assembly competition. However, once recruited they would disengage more slowly because they integrate principal cell firing over a longer time frame. In contrast, PV+ cell firing is thought to define the period for gamma oscillations. As a result, NMDARs on PV+ interneurons are ideally suited to allow for stable assemblies while the stimulus is constant, but also allow for quick switching if the nature of the stimulus changes.

Recurrent connections between PV+ interneurons and local pyramidal cell circuits are found throughout the brain. A challenge for future studies will be to establish whether feedback connections outside of hippocampal area CA1 also display NMDAR-dependent supralinear feedback integration. Furthermore, experiments acutely and specifically blocking NMDARs at feedback synapses onto PV+ cells, as opposed to nonspecific knockdown of all PV+ cell NMDARs during development, will be necessary in order to fully characterize the importance of PV+ cell NMDAR processing in vivo. Taken together, our results expand the computational role of NMDARs on PV+ cells, providing a parsimonious mechanism uniting a number of hitherto unexplained observations, relating to both basic neuronal network function and pathophysiology.

# Materials and methods

**Key resources table**

| Reagent type (species) or resource | Designation | Source or reference | Identifiers | Additional information |
|---|---|---|---|---|
| Strain, strain background (*Mus musculus*) | B6;129P2-*Pvalb*$^{tm1(cre)Arbr}$/J | The Jackson Laboratory | 008069 | |
| Strain, strain background (*Mus musculus*) | B6.Cg-*Gt(ROSA)26Sort m9(CAG-tdTomato)Hze* | The Jackson Laboratory | 007909 | |
| Other | AAV5-CaMKIIa-h ChR2(H134R)-EYFP | UNC Vector Core | | |
| Chemical compound, drug | Picrotoxin | Sigma-Aldrich | P1675 | |
| Chemical compound, drug | CGP 55845 | Abcam | Ab120337 | |
| Chemical compound, drug | D-AP5 | Tocris | 0105 | |
| Chemical compound, drug | TTX | Tocris | 1078 | |
| Chemical compound, drug | ZD 7288 | Tocris | 1000 | |
| Chemical compound, drug | MNI-glutamate TFA | Femtonics | 1951 | |

## Animals

Hippocampal slices were obtained from postnatal day 14–24 male and female mice, or from 2 to 3 month old male and female mice (optogenetic experiments), expressing tdTomato in PV+ interneurons. Experimental mice were obtained by crossing homozygous mice expressing Cre under the PV promoter (Jackson Labs: B6;129P2-*Pvalbtm1(cre)Arbr*) with homozygous Ai9 Cre reporter mice (Jackson Labs: B6.Cg-*Gt(ROSA)26Sortm9(CAG-tdTomato)Hze*). Animals were group-housed under a non-reversed 12 hr light/dark cycle, and allowed access to food and water ad libitum. All procedures were carried out in accordance with the UK Animals (Scientific Procedures) Act, 1986.

## Surgery for viral injections

Mice (minimum age: 6 weeks) were anesthetized with isoflurane and virus (AAV5-CaMKIIa-hChR2 (H134R)-EYFP) was stereotaxically injected into the dorsal CA1 region of both hippocampi using a Hamilton syringe. The injection coordinates were 2.15 mm caudal and 1.4 mm lateral of Bregma, and 1.2 and 1.0 mm deep from the pia. 50 nl of virus was injected at each site at a rate of 100 nl/min, and the needle was left in place for 5 min following injections before withdrawal. Slices were prepared for experiments after a minimum of three weeks post-surgery.

## Slice preparation and electrophysiology

Acute sagittal brain slices (300 µm) were prepared using a Vibratome (Leica VT1200 S). Slices were cut in an ice-cold artificial cerebrospinal fluid (ACSF) solution, containing (in mM): NaCl (119), KCl (2.5), $NaH_2PO_4$ (1.25), $NaHCO_3$ (25), glucose (20), $CaCl_2$ (1.5), $MgSO_4$ (1.3), and saturated with 95% $O_2$, 5% $CO_2$. Slices were allowed to recover at 32 ˚C for 15 min after slicing, before subsequent storage in ACSF at room temperature. Older mice (>1 month) were transcardially perfused with ice-cold sucrose-based ACSF solution, containing (in mM): sucrose (75), NaCl (87), KCl (2.5), $NaH_2PO_4$ (1.25), $NaHCO_3$ (25), glucose (25), $CaCl_2$ (0.5), $MgCl_2$ (7), and saturated with 95% $O_2$, 5% $CO_2$. Slices were cut in the same solution, and left to recover at 32 ˚C for 15 min, before being transferred to normal ACSF (same as above but with 2.5 mM $CaCl_2$) for storage at room temperature. All experiments were carried out in ACSF maintained at 30 ˚ – 32 ˚C and perfused at 2–3 ml/min. Recordings were made from dorsal hippocampal slices. For dissection of AMPAR and NMDAR components at feed-forward and feedback synapses (*Figure 3A,B*) a modified ACSF containing 0.1 mM $MgSO_4$ was used to partially relieve $Mg^{2+}$ blockade of NMDARs at rest. NMDARs and AMPARs were sequentially blocked with D-AP5 (100 µM) and 2,3-dihydroxy-6-nitro-7-sulfamoyl-benzo[f]quinoxaline-2,3-dione (NBQX, 10 µM), respectively. Picrotoxin (100 µM) and CGP 55845 (1 µM) were included throughout these experiments, as well as the HCN channel blocker ZD 7288 (30 µM), which was included in order to hyperpolarize pyramidal cells and decrease network excitability. Picrotoxin (100 µM) and

CGP 55845 (1 μM) were also included throughout the optogenetic experiments (*Figure 4*). For blockade of NMDARs or Na$^+$ channels during uncaging experiments (*Figure 2*) D-AP5 (100 μM) or TTX (0.1 μM) were used, respectively.

Fluorescence-guided somatic whole-cell recordings were obtained from PV+ interneurons and pyramidal cells using a Multiclamp 700B amplifier (Molecular Devices), filtered at 5 kHz, and digitized at 20 kHz (National Instruments PCI-6221), with LabVIEW Virtual Instruments. Patch pipettes of 3–4 MΩ resistance were filled with KGluconate- or CsGluconate-based internal solution for current clamp or voltage clamp experiments, respectively. These solutions contained (in mM): KGluconate (140), KOH-HEPES (10), EGTA (0.2), NaCl (8), Mg-ATP (2), Na-GTP (0.3) or CsGluconate (125), HCsO-HEPES (10), EGTA (0.2), NaCl (8), Mg-ATP (4), Na-GTP (0.33), Na-Phosphocreatine (10), TEA-Cl (5) QX314 (5). Cells were held at −60 mV or +60 mV during voltage-clamp experiments (*Figure 3A,B*, and *Figure 4B,C*). All other experiments were performed in current-clamp mode, and current was continuously injected to maintain cell membrane between −65 and −70 mV. The series resistance during voltage clamp recordings was <15 MΩ and during current clamp recordings was <25 MΩ. For field stimulation experiments, concentric bipolar stimulating electrodes (FHC) coupled to constant current stimulators (Digitimer) were placed in the alveus and stratum radiatum to evoke responses from feedback and feedforward inputs, respectively (*Pouille and Scanziani, 2004*). Stimuli were delivered to each pathway at 0.05 Hz, and alternated between the pathways. Optogenetic responses were elicited using 470 nm light pulses (1 ms, 1–15 mW) at 0.2 Hz, generated by an LED light-source (ThorLabs) and delivered through a 40X objective lens (Olympus). The light power necessary to elicit maximal and minimal responses was identified for each cell, and the difference in power was divided by five to define the five stimulation strengths (20%, 40%, 60%, 80% and 100%) used in each cycle. Maximal responses were identified as the maximum response elicited without generating an action potential (typically ≤15 mV), and minimal responses were the smallest response visible.

## Two-photon imaging and uncaging experiments

Slices were submerged in a perfusion chamber on an upright microscope (FV1000 BX61, Olympus). Simultaneous two-photon imaging and uncaging of MNI-caged glutamate was performed with two Ti-sapphire lasers tuned to 810 nm and 720 nm, for imaging and uncaging respectively (Mai-Tai, Spectra Physics; Chameleon, Coherent). MNI-caged-glutamate-TFA (3 mM; Femtonics) dissolved in the recording ACSF solution was perfused in a closed system.

Uncaging locations (range: 8–12, mean: 8.4 vs 8.5 sites in radiatum and oriens respectively) were selected either side of a dendritic region of interest, separated by 2–3 μm and within 1 μm of the dendrite. Uncaging-evoked EPSPs (uEPSPs) were evoked using 0.5 ms-long pulses of 720 nm laser-light. To account for differing depths of dendritic segments, uncaging-laser intensity was adjusted using a Pockels Cell (Conoptics). uEPSPs were first evoked by sequential stimulation of individual uncaging spots with an inter-stimulus interval of 200 ms. Uncaging locations (order chosen at random) were then stimulated at intervals of 1 ms, with 10 s delays between trials. Beginning with a single location, the number of uncaging locations was increased on successive trials until all locations were activated. The entire sequence was repeated between 2 and 5 times, without changing the order of uncaging locations, and the responses to each combination of uncaging locations were averaged. Arithmetic compound uEPSPs were constructed offline from the average of 5–8 responses to uncaging at each individual location, including a 1 ms waveform shift to match the experimental protocol, and compared to recorded uEPSPs. Uncaging times and locations were controlled by scanning software (Fluoview 1000V) and a pulse generator (Berkeley Nucleonics) coupled to the Pockels cell. Experiments were discontinued and excluded from analysis if photo-damage to PV+ cells was observed, or if physical drift occurred.

For experiments in which uncaging locations were placed across pairs of dendrites, uEPSPs were elicited from 12 locations in total, six on each dendrite. Glutamate was uncaged at locations across the two dendrites in five distinct patterns. Patterns 1 and 2 were on single dendrites (using alternate uncaging locations 1, 3, …,11 and 2, 4, …, 12), whereas patterns 3–5 ('mixed' dendrites) were across both dendrites (uncaging locations 1, 2, 3, 4, 5, 6; 4, 5, 6, 7, 8, 9; and 7, 8, 9, 10, 11, 12). For all patterns, locations on the same dendrite were activated with an interval of 2 ms, and waveforms were averaged across four repetitions. We constructed arithmetic compound uEPSPs for each pattern offline from the average of approximately eight responses to uncaging at each individual

location as above, and calculated the amplitude nonlinearity of the recorded vs arithmetic uEPSP. To compare between uESPSs evoked on single vs mixed dendrites we averaged the amplitude nonlinearity from patterns 1–2 and 3–5, respectively.

## Quantification and statistical analysis

Data analysis was performed using custom code written in Python. The nonlinearity of responses recorded from uncaging at each dendritic location was quantified using the following equation:

$$\% \ nonlinearity = \sum_{i=2}^{n} \frac{\frac{Mi}{Ai} - 1}{n - 1} \cdot 100\% \tag{1}$$

where $Mi$ is the amplitude of the $ith$ measured uEPSP (composed of $i$ individual uncaging spots), $Ai$ is the amplitude of the $ith$ constructed arithmetic summed uEPSP, and $n$ is the total number of uncaging locations. For uEPSP integral analysis a Savitzky-Golay filter was applied to traces.

For analysis of NMDAR/AMPAR ratios in *Figure 3A*, the AMPAR-mediated response was calculated by subtracting the NMDAR-mediated response (recorded in NBQX) from the baseline EPSC. NMDAR and AMPAR charge were then calculated by integrating the first 500 ms or 20 ms, respectively, of these isolated traces. NMDAR/AMPAR ratios in *Figure 4C* were calculated as the ratio between the NMDAR-mediated response recorded at +60 mV in the presence of NBQX (with the +60 mV response in the presence of D-AP5 subtracted), and the AMPAR-mediated response recorded at −60 mV.

Statistical significance was assessed using Student's paired or unpaired *t*-tests. Data are presented as mean ± SEM, unless stated otherwise. Sample sizes were estimated to obtain 80% power to detect effects at p<0.05. *n* values are for cells.

## Multi-compartmental modeling

Multi-compartmental modeling was performed with the NEURON 7.5 simulation environment (*Hines and Carnevale, 1997*). The soma and dendrites of a PV+ interneuron were reconstructed using the TREES toolbox in MATLAB (*Cuntz et al., 2010*). The axon was not included in the reconstruction. As PV+ interneuron dendrites are generally smooth, addition of spines or correction of synaptic responses for spines was deemed unnecessary. The number of segments per section was constrained to odd numbers and set according to the d-lambda rule (*Carnevale and Hines, 2009*) to have a length no more than 10% of the alternating current length constant at 1 kHz. The model contained 500 segments in total with a maximal segment length of 8.7 µm.

The biophysical parameters were based on previously published models of dentate gyrus PV+ basket cells (*Hu and Jonas, 2014*; *Nörenberg et al., 2010*). The specific membrane capacitance ($C_m$) and intracellular resistance ($R_i$) were assumed to be spatially uniform (for values see *Table 1*). In contrast, the specific membrane resistance ($R_m$) was assumed to vary as a step function with distance from the soma. $R_m$ at distal dendrites was 10 times larger than at proximal dendrites, and $R_m$ was

**Table 1.** NEURON model parameters.

| Parameter | Proximal | Distal | Units |
|---|---|---|---|
| $C_m$ | 0.9 | 0.9 | µF cm$^{-2}$ |
| $R_{axial}$ | 170 | 170 | Ω cm |
| $R_m$ | 5.55 | 55.5 | kΩ |
| $e_{leak}$ | -65 | - | mV |
| $e_{gk}$ | -90 | - | mV |
| $e_{gNa}$ | 55 | - | mV |
| v Shift | -12 | -12 | mV |
| $g_k$ dend | 300 | 300 | pS µm$^{-2}$ |
| $g_{Na}$ dend | 200 | 100 | pS µm$^{-2}$ |
| $g_{Na}$ Soma | 2000 | - | pS µm$^{-2}$ |

DOI: https://doi.org/10.7554/eLife.49872.020

chosen so as to make the model cell's input resistance 78 MΩ, close to the average experimentally recorded input resistance (78.6 ± 5.2 MΩ). The border between proximal and distal dendrites was defined to be 120 μm from the soma.

Wang and Buzsaki (WB) Na$^+$ and K$^+$ channels were inserted in the model neuron to confer a fast-spiking action potential phenotype (*Wang and Buzsáki, 1996*). However, in order to produce a realistic firing frequency – current injection relationship, a hyperpolarizing voltage shift was included in the WB implementation. The depolarized threshold of the WB mechanism has been discussed previously (*Ferguson et al., 2013*).

Subthreshold synaptic integration curves were produced by first finding all sites on the dendritic tree that were located between 40 and 190 μm from the soma. Simulations then closely followed the experimental protocol detailed above. At each dendritic site, 15 synapses were placed within a distance of 30 μm. Each synapse was activated individually and the arithmetic sum calculated from the somatic membrane potential. Synapses were then activated in increasing numbers, with an interval of 1 ms between activations, and the integral and amplitude of these measured responses compared to the calculated arithmetic responses. Quantification of dendritic nonlinearity was identical to that applied to experimental data.

## Network modeling
### Single cell modeling
For network simulations, PV+ interneurons and CA1 pyramidal cells were represented by two-dimensional Izhikevich model neurons (*Izhikevich, 2003*). Izhikevich models for these neurons have previously been parameterized from experimental data (*Ferguson et al., 2014*; *Ferguson et al., 2013*). In line with this previous work, the models were slightly modified to reproduce the narrow PV+ interneuron spike width, and had the following form:

$$C_m \frac{dv}{dt} = k(v - v_r)(v - v_t) - u + I_{applied}$$
$$\frac{du}{dt} = a[b(v - v_r) - u]$$

(2)

$$\text{if } v \geq v_{peak}, \text{ set } v = c, \ u = u + d$$

$$\text{where } k = k_{low} \text{ if } v \leq v_t; \ k = k_{high} \text{ if } v_t$$

The variable $v$ represents the membrane potential, and $u$ represents a slow 'refractory' current, that, aside from subthreshold effects governed by $b$, is increased by $d$ when the neuron fires, and decays at a rate determined by $a$. The parameter $C_m$ represents the membrane capacitance; $k$ is a scalar; $v_r$ is the resting membrane potential; $v_t$ is the instantaneous spiking threshold potential; $v_{peak}$ is the peak action potential voltage; $I_{applied}$ is the applied current, comprised of the sum of all synaptic inputs to the cell; $a$ is the recovery inverse time constant of the refractory current, u; $b$ is the sensitivity of $u$ to subthreshold voltage fluctuations; $c$ is the voltage reset value; and $d$ is the amount of current generated by the after-spike behavior. The values for all parameters of the network simulations are presented in *Table 2*.

### Synaptic modeling
Synaptic connections between neurons were modeled as bi-exponential, conductance-based synapses (*Roth and van Rossum, 2009*), which can be written in the following differential form:

$$\begin{aligned}
g^{chan}(v^{syn}) &= \frac{G^{chan}(v^{syn})}{\tau_1^{chan} - \tau_2^{chan}}\left(g_1^{chan} - g_2^{chan}\right) \\
\frac{dg_1^{chan}}{dt} &= \left(\delta^{chan}(t) - \frac{1}{\tau_1^{chan}} g_1^{chan}\right) \\
\frac{dg_2^{chan}}{dt} &= \left(\delta^{chan}(t) - \frac{2}{\tau_2^{chan}} g_2^{chan}\right) \\
G^{NMDA}(v^{syn}) &= \left(\frac{1}{2} tanh\left[\frac{v_i^{syn} + 50mV}{10mV}\right] + \frac{1}{2}\right) \\
G^{AMPA}(v^{syn}) &= G^{ext}(v^{syn}) = G^{GABA}(v^{syn}) = 1
\end{aligned}$$

(3)

**Table 2.** model neuron parameters for network modeling.

| Parameter | FS PV+ | Pyramidal | Units |
|---|---|---|---|
| $C_m$ | 90 | 115 | pF |
| $k_{low}$ | 1.7 | 0.1 | nS/mV |
| $k_{high}$ | 14 | 3.3 | nS/mV |
| $v_r$ | −60.6 | −65.8 | mV |
| $v_{peak}$ | 2.5 | 22.6 | mV |
| $v_t$ | −43.1 | −57 | mV |
| $a$ | 0.1 | 0.0012 | ms$^{-1}$ |
| $b$ | −0.1 | 3 | nS |
| $c$ | −67 | −65.8 | mV |
| $d$ | 0.1 | 10 | pA |
| $\tau_1^{AMPA}$ | 0.25 | 0.2 | ms |
| $\tau_2^{AMPA}$ | 0.77 | 1.7 | ms |
| $\tau_1^{GABA}$ | 0.27 | 0.3 | ms |
| $\tau_2^{GABA}$ | 1.7 | 3.5 | ms |
| $\tau_1^{NMDA}$ | 2 | - | ms |
| $\tau_2^{NMDA}$ | 60 | - | ms |
| $e^{glu}$ | 0 | 0 | mV |
| $e^{GABA}$ | −70 | −70 | mV |
| $\sigma_D^2$ | $0.015/(n_{pyr}^2)$ | - | - |
| $C_{syn}$ | 9 | - | pF |
| $k^{syn}$ | $3/n_{pyr}$ | - | - |
| $g^{leak}$ | 5 | - | nS |
| $e^{leak}$ | −60.6 | - | mV |
| $k^{AMPA}$ | $2^8$ | - | - |
| $k^{NMDA}$ | $2^{12}$ | - | - |
| $k^{GABA}$ | $2^8$ | $2^7$ | - |
| $k^{ext}$ | 5 | 1 | - |

Izhikevich Parameter values as in *Ferguson et al. (2014)* and *Ferguson et al. (2013)*. Synaptic time constants: (*Bartos et al., 2002*; *Geiger et al., 1997*; *Roth and van Rossum, 2009*). Remaining parameters where adjusted to allow the network to generate a gamma rhythm.

DOI: https://doi.org/10.7554/eLife.49872.021

Where $g^{chan}$ is the total synaptic conductance of a given channel family (composed of a rise term $g_1^{chan}$ and a decay term $g_2^{chan}$), $G^{chan}(v^{syn})$ is an eventual instantaneous voltage-gating term dependent on a local synaptic voltage $v^{syn}$ (only relevant for NMDA channels; detailed in the next section), $\tau_1^{chan}$ is the rising exponential time constant, $\tau_2^{chan}$ is the decay exponential time constant, the variable $t$ is time. The function $\delta^{chan}(t)$ represents the input spike train. This is defined in continuous time in order to be agnostic to the numerical integration method used in the simulations. Specifically $\delta^{chan}(t)$ is modeled as:

$$\delta^{chan}(t) = \sum_i^{Nspikes} \delta(t - t_i) \tag{4}$$

Where $t_i$ are the times of the input spike arrivals for a particular receptor, and $\delta(t)$ is the continuous time Dirac delta distribution (0 everywhere apart from t=0 where it has infinite density and integrates to 1).

## NMDA receptor modeling

NMDARs, present at the feedback connections from principal cells onto the interneuron, and the cooperative relief of NMDAR $Mg^{2+}$ block by co-active synaptic inputs, were modeled in an abstract manner. We assumed that all co-active inputs from the population of $n_{pyr}$ principal cells had a degree of cooperation, or functional clustering (*Wilson et al., 2016*), which was weighted by a distance matrix, $D^{n_{pyr} \times n_{pyr}}$. $D$ was defined as a Toeplitz matrix, with the $i$th element of the row vector $D_{\frac{n_{pyr}}{2}}$ equal to:

$$D_{\frac{n_{pyr}}{2}, \, i} = \frac{1}{\sqrt{2\pi \, \sigma_D^2}} e^{-\frac{(i - n_{pyr}/2)^2}{2\sigma_D^2}} \tag{5}$$

Where $\sigma_D^2$ controlled the specificity of local cooperation. We then modeled the time evolution of the voltage of the local synaptic membrane patch, $v_i^{syn}$, with the following equation:

$$
\begin{aligned}
C_{syn} \frac{dv_i^{syn}}{dt} &= k^{syn} I_i^{syn} + g^{leak}\left(e^{leak} - v_i^{syn}\right) \\
I_i^{syn} &= \sum_{j=1}^{n_{pyr}} D_{i,j}\left(k_{pv+}^{AMPA} g_j^{AMPA} + k_{pv+}^{NMDA} g_j^{NMDA}(v_j^{syn})\right)\left(e^{ghu} - v_i^{syn}\right)
\end{aligned} \tag{6}
$$

Where $C_{syn}$ is the local patch membrane capacitance, $k^{syn}$ is a gain applied to the input current $I_i^{syn}$, $k_{pv+}^{chan}$ are gain constants defining each channel family synaptic strength onto the interneuron, $g^{leak}$ is a leak conductance with reversal potential $e^{leak}$.

Finally, the total current, $I_{applied}$, an interneuron receives is given by:

$$
\begin{aligned}
I_{applied} = \; &\left(\sum_{i=1}^{n_{pyr}} k_{pv+}^{AMPA} g_j^{AMPA} + k_{pv+}^{NMDA} g_j^{NMDA}(v_i^{syn}) + k_{pv+}^{ext} g_i^{ext}\right)\left(e^{glu} - v\right) \\
&+ k_{pv+}^{GABA}\left(\sum_{j=1}^{n_{pv+}} g_j^{GABA}\right)\left(e^{GABA} - v\right)
\end{aligned} \tag{7}
$$

Accordingly, for the $i$th pyramidal cell the input current is more simply given by:

$$I_{Applied}^i = k_{pyr}^{ext} g_i^{ext}(e^{glu} - v) + k_{pyr}^{GABA}\left(\sum_{j=1}^{n_{pv+}} g_j^{GABA}\right)(e^{GABA} - v) \tag{8}$$

## External drive

The external drive to the $i$th pyramidal neuron was modeled as a non-homogeneous Poisson process, in which the instantaneous rate of incoming spikes was given by an Ornstein–Uhlenbeck process with mean $\mu_{OU}$, volatility $\sigma_{OU}$ and time constant $\tau_{OU}$, multiplied by a Gaussian gain function representing the neuron position in the receptive field. The input of the PV+ interneurons was the scaled mean of all inputs to its afferent pyramidal cells. The peak mean $\mu_{OU}$ was 5000 spikes per second, $\tau_{OU}$ was 50 ms and $\sigma_{OU}$ was $\frac{1}{6}\mu_{OU}\sqrt{2\,\tau_{OU}}$ for the simulations in *Figure 7* and zero everywhere else (i.e. the Poisson rates were constant). These inputs are convolved with the AMPA synaptic conductance kernel defined in *Equation (3)* and parameterized by rise and decay time constants shown in *Table 2*.

## Acknowledgements

We are grateful to members of the Experimental Epilepsy Group at the UCL Institute of Neurology, in particular Kaiyu Zheng, and also to Peter Latham and Arnd Roth for advice and technical assistance. This work was supported by the Wellcome Trust, the Medical Research Council, Epilepsy Research UK, and the Brain Research Trust.

## Additional information

### Funding

| Funder | Grant reference number | Author |
|---|---|---|
| Wellcome | 095580/Z/11/Z | Jonathan H Cornford<br>Marion S Mercier<br>Marco Leite<br>Vincent Magloire<br>Dimitri Michael Kullmann |
| Wellcome | 212285/Z/18/Z | Dimitri Michael Kullmann |
| Medical Research Council | MR/L01095X/1 | Vincent Magloire |
| Brain Research Trust | | Jonathan H Cornford |
| Wellcome | 201225/Z/16/Z | Michael Häusser |
| Epilepsy Research UK | P1702 | Vincent Magloire |
| ERC | AdG 695709 | Michael Häusser |

The funders had no role in study design, data collection and interpretation, or the decision to submit the work for publication.

### Author contributions

Jonathan H Cornford, Conceptualization, Data curation, Formal analysis, Investigation, Methodology, Writing—original draft, Writing—review and editing; Marion S Mercier, Conceptualization, Formal analysis, Investigation, Visualization, Writing—original draft, Writing—review and editing; Marco Leite, Conceptualization, Software, Formal analysis, Writing—review and editing; Vincent Magloire, Investigation, Methodology, Writing—review and editing; Michael Häusser, Supervision, Methodology, Writing—review and editing; Dimitri M Kullmann, Conceptualization, Formal analysis, Supervision, Validation, Methodology, Writing—original draft, Project administration, Writing—review and editing

### Author ORCIDs

Marion S Mercier https://orcid.org/0000-0003-3929-8118
Dimitri M Kullmann https://orcid.org/0000-0001-6696-3545

### Ethics

Animal experimentation: The study was performed in accordance with the Animals (Scientific Procedures) Act 1986 and reviewed by the Animal Welfare and Ethical Review Body (AWERB) of the UCL Queen Square Institute of Neurology.

### Decision letter and Author response

Decision letter https://doi.org/10.7554/eLife.49872.024
Author response https://doi.org/10.7554/eLife.49872.025

## Additional files

### Supplementary files

• Transparent reporting form DOI: https://doi.org/10.7554/eLife.49872.022

### Data availability

All data generated or analysed during this study are included in the manuscript and supporting files.

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
