## [Decision Letter]

**Acceptance summary:**

This study provides evidence that dendritic NMDA receptors underlie supralinear integration of excitatory feedback inputs from local principal cells onto CA1 PV-expressing fast-spiking interneurons. By using computational neuronal network models, this work further demonstrates that NMDA receptors in PV-interneurons support their cooperative recruitment and strengthening of principal cell assemblies.

**Decision letter after peer review:**

Thank you for submitting your article "Dendritic NMDA receptors in parvalbumin neurons enable strong and stable neuronal assemblies" for consideration by *eLife*. Your article has been reviewed by three peer reviewers, one of whom is a member of our Board of Reviewing Editors, and the evaluation has been overseen by Gary Westbrook as the Senior Editor. The following individuals involved in review of your submission have agreed to reveal their identity: Cheng-Chang Lien (Reviewer #2); Jean Christophe Poncer (Reviewer #3). The reviewers have discussed the reviews with one another and the Reviewing Editor has drafted this decision to help you prepare a revised submission.

Summary:

This manuscript provides a computational explanation on the potential role of NMDAR-mediated currents in PV interneurons in neuronal representation of given information. They show that dendrites receiving feedback excitation from local CA1 principal cells (PCS) show supralinear summation of EPSPs generated in close proximity at a dendritic compartment using glutamate uncaging. The data are reproduced by ChR2-mediated excitation of PC inputs. The authors apply a computational model consisting of PV cells and PCs and show that clustered synchronized excitatory inputs recruit PV cells if NMDAR-mediated EPSPs are induced. In contrast, randomized non-synchronized excitatory inputs do not reliably activate PV cells. By using neural network simulations, the authors provide evidence suggesting NMDAR enrichment may influence the properties of neuronal ensembles by promoting output/input fidelity and the stability of cell assemblies.

Essential revisions:

All three reviewers judge the work as high-quality, which provides new insights on the synaptic integration of excitatory inputs along the dendrites of parvalbumin (PV)-expressing interneurons. However, all reviewers formulated major criticisms, which need to be addressed by the authors.

1) The reviewers agree that the supralinear summation of uEPSPs evoked in PV+ interneuron dendrites in stratum oriens but not in stratum radiatum is central to the study, including the modeling part, but requires further investigation. Particularly, it is important to test the spatial constraints for supralinearity: how does spacing between simultaneously active inputs influence their summation and what happens when uEPSCs are evoked onto 2 distinct, same order sister dendrites? How does the number and duration of individual uncage locations influence the summation of EPSPs?

2) Figure 3—figure supplement 1 is important as it addresses whether the properties of GluA2-lacking AMPARs in PV+ cells influence synaptic cooperativity. It would be perhaps more explicit to represent these data as in Figure 3D-E and include the appropriate statistics.

3) State clearly that uncaging of glutamate does not necessarily activate synaptic receptors. See statement of reviewer #3 point 2 for details.

4) Figure 5—figure supplement 2 is important, as it describes the effect of NMDAR enrichment on the sharpness of the network response to a clustered burst of input activity. However, in its present form it is only descriptive. Can these simulation data be tested for statistically significant difference? Moreover, explain on what ground they chose the first and last 75 ms of network simulation, as it is not clear from Figure 5D that networks dynamics evolve much over time. It would be useful if the authors could include a panel similar to the middle panel in Figure 5D for the case of no NMDARs.

5) Excitatory inputs: the authors model external inputs to pyramidal neurons using a Poisson process. Although this is often done for modeling cortical networks, the reviewer is unsure whether this is most adapted to area CA1 of the hippocampus. It would perhaps be more informative to use Gaussian-modulated theta-frequency input. This would also let the authors test how/whether γ-band firing of PV cells is modulated by theta-modulated inputs. Moreover, supralinearity of excitatory inputs in this study is dependent on spatial clustering of glutamatergic inputs. Please discuss on what published information is available on the spatial distribution of co-active inputs onto CA1-PV interneuron dendrites in vivo.

6) There is always a concern about 'tuning' models to fit the desired output. Most modeling studies therefore perform a sensitivity analysis of the various parameters that were fixed for the simulations (see Marder and Taylor 2011 Nat Neuroscience; Rathour and Narayanan 2014 PNAS). Was such an analysis conducted within this study to ensure that no bias was generated in the network output due to the choice of various parametric values? Comment on this point in the manuscript.

7) AAV-ChR2 was injected in the dorsal hippocampus. Please comment on what part of the hippocampus has been used for in vitro whole-cell recordings in the study.

---

## [Author Response]

Essential revisions:All three reviewers judge the work as high-quality, which provides new insights on the synaptic integration of excitatory inputs along the dendrites of parvalbumin (PV)-expressing interneurons. However, all reviewers formulated major criticisms, which need to be addressed by the authors.1) The reviewers agree that the supralinear summation of uEPSPs evoked in PV+ interneuron dendrites in stratum oriens but not in stratum radiatum is central to the study, including the modeling part, but requires further investigation. Particularly, it is important to test the spatial constraints for supralinearity: how does spacing between simultaneously active inputs influence their summation and what happens when uEPSCs are evoked onto 2 distinct, same order sister dendrites? How does the number and duration of individual uncage locations influence the summation of EPSPs?

We agree that the experimental parameters used in the glutamate uncaging experiments are a subset of the possible range of conditions that could potentially be investigated. Our network simulations rely on the assumption that some dendritic inputs can co-operate while others do not, because they are electrically too remote from one another to allow synaptic depolarization to facilitate the opening of NMDARs. We therefore carried out a set of experiments in which we compared the supralinear summation of uEPSPs when uncaging was either performed within dendrites or across two dendrites. In line with expectation, the degree of supralinearity was significantly greater when the uncaging locations were placed on single dendrites than when distributed across two dendrites. This provides further evidence that the supralinearity results from the spread of depolarization within dendrite branches, facilitating the opening of NMDARs. We should point out that the experiment is not trivial, because it requires two dendrites to be identified in the same optical plane, and oriented in such a way that it is possible to position multiple uncaging locations. This meant that we were unable to ensure in all cases that the uncaging was on the same order branches. We have added a new figure (Figure 1—figure supplement 5) to report the results.

As for the effect of increasing the number of individual uncaging locations, this is shown in Figure 1E, F and Figure 2D. Biophysical modelling (Figure 3) predicts that the supralinearity should saturate with very large inputs and there is a suggestion of a sigmoid relationship in the experimental data but the roll-off is too subtle to test statistically. With respect to uncaging duration, we have not explored durations shorter than 0.5 ms because we cannot confine the uncaging to a spatial scale below 1 μm, which very roughly corresponds to a diffusional delay of 1 ms. Conversely, longer uncaging durations would not be representative of synaptic glutamate release.

2) Figure 3—figure supplement 1 is important as it addresses whether the properties of GluA2-lacking AMPARs in PV+ cells influence synaptic cooperativity. It would be perhaps more explicit to represent these data as in Figure 3D-E and include the appropriate statistics.

Agreed and done.

3) State clearly that uncaging of glutamate does not necessarily activate synaptic receptors. See statement of reviewer #3 point 2 for details.

We now stress this (subsection “NMDAR recruitment at CA1 pyramidal cell feedback connections onto PV+ interneurons”, second paragraph).

4) Figure 5—figure supplement 2 is important, as it describes the effect of NMDAR enrichment on the sharpness of the network response to a clustered burst of input activity. However, in its present form it is only descriptive. Can these simulation data be tested for statistically significant difference? Moreover, explain on what ground they chose the first and last 75 ms of network simulation, as it is not clear from Figure 5D that networks dynamics evolve much over time. It would be useful if the authors could include a panel similar to the middle panel in Figure 5D for the case of no NMDARs.

We now show the profile for each cycle of the γ oscillation to illustrate the evolution of the behaviour of the network (Figure 5—figure supplement 2). We have repeated the simulations to test for robustness and have added confidence intervals. We hesitate to give p values for the differences because they can be reduced to arbitrarily low values by increasing the number of simulation runs. The effect sizes are more informative than p values.

*5) Excitatory inputs: the authors model external inputs to pyramidal neurons using a Poisson process. Although this is often done for modeling cortical networks, the reviewer is unsure whether this is most adapted to area CA1 of the hippocampus. It would perhaps be more informative to use Gaussian-modulated theta-frequency input. This would also let the authors test how/whether γ-band firing of PV cells is modulated by theta-modulated inputs. Moreover, supralinearity of excitatory inputs in this study is dependent on spatial clustering of glutamatergic inputs. Please discuss on what published information is available on the spatial distribution of co-active inputs onto CA1-PV interneuron dendrites* in vivo.

We have repeated simulations using a theta-modulated Poisson process to represent the external input. This does not change the conclusions (new Figure 7—figure supplement 1). As for the spatial distribution of co-active inputs on CA1-PV interneuron dendrites, we are not aware of a literature directly addressing this, although we hope that our study will stimulate efforts in this direction, just as the original reports on dendritic computations (Losonczy and Magee, 2006, Branco et al., 2010) spurred research on the organization of projections to individual dendritic branches in principal cells (Takahashi et al., Science. 2012 Jan 20;335(6066):353-6).

6) There is always a concern about 'tuning' models to fit the desired output. Most modeling studies therefore perform a sensitivity analysis of the various parameters that were fixed for the simulations (see Marder and Taylor 2011 Nat Neuroscience; Rathour and Narayanan 2014 PNAS). Was such an analysis conducted within this study to ensure that no bias was generated in the network output due to the choice of various parametric values? Comment on this point in the manuscript.

We agree that tuning the parameters of a model to obtain a desired output is a concern for all modelling. However, it is also true that most parameter combinations in neuronal modelling yield unrealistic behaviours. It is therefore essential, whenever possible, to constrain these parameters with information agnostic to the main claims of the model. In our case, we did this primarily by using published estimates for well-defined parameters such as channel kinetics and cellular parameters. We then tuned the remaining parameters such that the network behaves in a biophysically plausible γ oscillation regime in terms of cellular membrane currents and neuronal firing patterns. The most critical and unconstrained parameters of relative AMPAR and NMDAR conductances were indeed swept across a wide range of values (Figure 7D and Figure 7—figure supplement 1C). This was to ensure that the observed effect of altering NMDARs could not be explained by the alternate candidate (AMPARs) across a wide range of conditions. It is also noteworthy that we have repeated the simulations with different estimates of the kinetics of GABAergic conductances, and with different input patterns (in response to point 5 above) and, while the γ oscillation frequency changed as expected, the relative sensitivity of the flip rate of the bi-stable network to changes in AMPAR and NMDAR conductances was unaffected (Figure 7).

Finally, while the sensitivity analyses performed in the mentioned references are relatively simple to perform for quantitative parameter estimates in single cell models, they are much more computationally costly to perform in network simulations. Our conclusions rely more on the general qualitative behaviour of the model, rather than in its specific quantitatively derived estimates. Therefore, we judged the exhaustive sensitivity analysis of the remaining parameters to be less relevant for our application.

*7) AAV-ChR2 was injected in the dorsal hippocampus. Please comment on what part of the hippocampus has been used for* in vitro *whole-cell recordings in the study.*

Done.